# Multi-Sensor Adaptive Weighted Data Fusion Based on Biased Estimation

**DOI:** 10.3390/s24113275

**Published:** 2024-05-21

**Authors:** Mingwei Qiu, Bo Liu

**Affiliations:** School of Information and Intelligent Science and Technology, Hunan Agricultural University, Changsha 410127, China; wf2511574305@163.com

**Keywords:** adaptive weighted data fusion, unbiased estimation, biased estimation

## Abstract

In order to avoid the loss of optimality of the optimal weighting factor in some cases and to further reduce the estimation error of an unbiased estimator, a multi-sensor adaptive weighted data fusion algorithm based on biased estimation is proposed. First, it is proven that an unbiased estimator can further optimize estimation error, and the reasons for the loss of optimality of the optimal weighting factor are analyzed. Second, the method of constructing a biased estimation value by using an unbiased estimation value and calculating the optimal weighting factor by using estimation error is proposed. Finally, the performance of least squares estimation data fusion, batch estimation data fusion, and biased estimation data fusion is compared through simulation tests, and test results show that biased estimation data fusion has a greater advantage in accuracy, stability, and noise resistance.

## 1. Introduction

Data fusion can utilize the information of multiple different data sources to complement each other, so the application of data fusion algorithms can suppress negative environmental factors and get more accurate results [1,2]. Data fusion algorithms in combination with various algorithms can achieve better results [3,4], so they have been widely used in various fields [5,6]. However, data fusion can only be applied to linear systems where all noises are independent of each other; if it is applied to a nonlinear system, it needs to be combined with the extended Kalman filter or the unscented Kalman filter [7]. If noises are correlated, they need to be decoupled by linear changes [8].

It is necessary to first process the original data, which includes removing abnormal data, filtering system deviations, and lossless compression of data. Abnormal data is removed using the Grubbs criterion. Filtering bias uses bias estimation algorithms [9,10], and in distributed fusion, if data need to be restored in the data center, lossless compression is required [11].

Before conducting multi-sensor data fusion, each sensor can also use various algorithms to estimate the true value (called the estimation value), which can improve the fusion accuracy because the estimation value has higher accuracy than the measurement value.

When the measured physical quantity does not change over time, if there is sufficient understanding of the system, linear minimum mean square error estimation is the best method. This algorithm can find a linear function that minimizes mean square error through a single measurement value [12,13,14]. Otherwise, it is best to use least squares estimation [15,16] and batch estimation [17]. When the measured physical quantity changes over time, it can only be suitable to use Kalman filtering [18,19].

Adaptive weighted data fusion is a widely used multi-sensor data fusion algorithm with a unique optimal weighting factor that enables the fusion estimation value to have a lower estimation variance than the measurement variance of all sensors, but it requires prior knowledge of the measurement variance to run [20]. However, its combination with a partially unbiased estimation algorithm (hereinafter referred to as unbiased estimation data fusion) not only further reduces the estimation variance but also removes the need for prior knowledge.

Unbiased estimation data fusion has two shortcomings. The first one is that unbiased estimation does not mean that the estimation results must be reliable. According to the Gauss-Markov theorem, the unbiased estimation variance has a lower bound, and when the lower bound itself is very large, even if the unbiased estimation variance is minimized, the difference between the estimation results and the true value (hereinafter referred to as estimation error) may not be within the acceptable range [21]. The other one is that the optimal weighting factor is not necessarily optimal; the optimal weighting factor is calculated according to the measurement variance; a smaller measurement variance does not necessarily mean a smaller measurement error, so in some cases, the optimal weighting factor is not optimal. Therefore, this paper proposes a biased estimation data fusion algorithm that also does not require priori knowledge, which uses unbiased estimation values to construct biased estimation values with lower estimation errors and calculates the optimal weighting factor according to the estimation error to ensure its optimality.

This paper is organized as follows: Section 2 reviews some existing unbiased estimation data fusion algorithms, Section 3 describes the shortcomings of the existing algorithms and the feasibility of the improved methods, Section 4 describes the specific implementation of the algorithms in this paper, Section 5 compares the performance of the different algorithms, and Section 6 summarizes the entire content and provides prospects for the future.

## 2. Unbiased Estimation Data Fusion

When N sensors are measuring an invariant physical quantity and all the sensor nodes in a network measure the observed physical quantity at the same time, the measurement equation is:(1)$${z(k)}_{i}=H_{i}x+{v(k)}_{i}$$
where x is the unknown quantity to be measured, μ is the true value of x, and k is the discrete time indicator, ${z(k)}_{i}$ is the measurement value of the *i*th sensor at the moment k, $H_{i}$ is the measurement matrix of the *i*th sensor (the H are often unit matrices), ${v(k)}_{i}$ is the measurement noise, ${v(k)}_{i}\sim N(0,v_{i}^{2})$ and are independent of each other, and $v_{i}^{2}$ be called measurement variance. Moreover, according to the additivity of the normal distribution, ${z(k)}_{i}\sim N(\mu ,v_{i}^{2})$.

In this system, only ${z(k)}_{i}$ is a known quantity; how one can use ${z(k)}_{i}$ to calculate the true value of μ? Since the measurement equations are designed for multiple sensors and the measurement variances may vary, adaptive weighted data fusion is often used to solve such problems, but adaptive weighted data fusion needs prior knowledge of the measurement variance of all the sensors, and in practice, the measurement covariance can be obtained by experiments.

### 2.1. Unbiased Estimation

Unbiased estimation refers to: if the mathematical expectation of an estimator is equal to the true value of the estimated parameter (this property is called unbiasedness), then this estimator is called the unbiased estimation of the estimated parameter. The mathematical expectation of the estimator constructed by unbiased estimation is equal to the true value of the estimated parameter, so using unbiased estimators instead of measurement values will not cause an increase in error. Unbiased estimation of sample variance can be used to replace the measurement variance of sensors, allowing adaptive weighted data fusion to operate without knowing the measurement variance of sensors. There are two commonly used unbiased estimation algorithms, as follows:

Least squares estimation:

Assuming that a particular sensor collects n data over a period of time, $\bar{T}$, the estimation value, and $\hat{\sigma^{2}}$, estimation variance, are given as follows:(2)$$\bar{T}=\;\frac{1}{n}\;\sum_{i=1}^{n}T_{i}$$
(3)$$\hat{\sigma^{2}}=\frac{\sum_{i=1}^{n}{\left(T_{i}-\bar{T}\right)}^{2}}{n-1}$$

Batch estimation:

Assuming that a particular sensor collects n data over a period of time, these n data are randomly divided into two groups, where the data in the *i*th groups are: $T_{i1},T_{i2}\cdots T_{in_{i}}(\frac{n}{2}\geq n_{i}\geq 2)$, i=1,2. ${\bar{T}}_{i}$, the sample mean, and $\sigma_{i}^{2}$, sample variance, are given as follows:(4)$${\bar{T}}_{i}=\;\frac{1}{n_{i}}\;\sum_{j=1}^{n_{i}}T_{ij}\;\;\;\;i=1,\;2$$
(5)$$\sigma_{i}^{2}=\frac{\frac{1}{n_{i}-1}\sum_{j=1}^{n_{i}}{\left(T_{ij}-{\bar{T}}_{i}\right)}^{2}}{n_{i}}\;\;\;i=1,\;2$$

According to the theory of batch estimation in statistics, the optimal estimation value $\hat{T}$ and the optimal estimation variance $\hat{\sigma^{2}}$ of these 2 sets of data are given as follows:(6)$$\hat{T}=\frac{\sigma_{1}^{2}{\bar{T}}_{2}+\sigma_{2}^{2}{\bar{T}}_{1}}{\sum_{i=1}^{2}\sigma_{i}^{2}}$$
(7)$$\hat{\sigma^{2}}=\frac{\sigma_{1}^{2}{\;\sigma }_{2}^{2}}{\sum_{i=1}^{2}\sigma_{i}^{2}}$$

### 2.2. Adaptive Weighted Data Fusion

Adaptive weighted data fusion principle: assume that there are N sensors working on the same unknown quantity x (the true value is μ) is measured, and the measurement value of the *i*th sensor is $M_{i}$. The measurement variance of the *i*th sensor is $v_{i}^{2}$, and the measurement data of the *i*th sensor follows $N\left(\mu ,v_{i}^{2}\right)$, then the unbiased estimator $\hat{M}$ is given as follows:(8)$$\hat{M}=\sum_{i=1}^{N}w_{i}M_{i}$$
(9)$$\hat{\sigma^{2}}=\sum_{i=1}^{N}w_{i}^{2}v_{i}^{2}$$

Since $\hat{M}$ is an unbiased estimator, so that
(10)$$\sum_{i=1}^{N}w_{i}=1$$

In order to find $w_{i}$ that minimizes $\hat{\sigma^{2}}$ (call this $w_{i}$ as the optimal weighting factor), the auxiliary function that is constructed according to the Lagrange multiplier method is given as follows:(11)$$f\left(w_{1},w_{2},\cdots ,w_{n},\varphi \right)=\sum_{i=1}^{N}w_{i}^{2}v_{i}^{2}+\varphi (\;1-\sum_{i=1}^{N}w_{i})$$

According to multivariate extremum theory, it is known that the solution of the following partial differential equation makes the estimation variance $\hat{\sigma^{2}}$ minimized, as follows:(12)$$\left\{\begin{array}{c} \frac{\partial f}{\partial w_{1}}=2w_{1}v_{1}^{2}-\varphi =0\\ \cdots \\ \frac{\partial f}{\partial w_{n}}=2w_{n}v_{n}^{2}-\varphi =0\\ \;\frac{\partial f}{\partial \varphi }=\left(\;1-\sum_{i=1}^{N}w_{i}\right)=0 \end{array}\right.$$

The simplified formula is given as follows:(13)$$\left\{\begin{array}{c} w_{i}=\frac{\theta }{v_{i}^{2}},\;i=1,2,\cdots ,n;\theta =\frac{\varphi }{2}\\ \sum_{i=1}^{N}w_{i}=1 \end{array}\right.$$

Thus, the formula for $w_{i}$ are given as follows:(14)$$w_{i}=(v_{i}^{2}\sum_{i=1}^{n}\frac{1}{v_{i}^{2}}{)}^{-1}$$

So, $\hat{M}$ and $\hat{\sigma^{2}}$ are given as follows:(15)$$\hat{M}=\sum_{i=1}^{N}(v_{i}^{2}\sum_{i=1}^{n}\frac{1}{v_{i}^{2}}{)}^{-1}M_{i}$$
(16)$$\hat{\sigma^{2}}=(\sum_{i=1}^{n}\frac{1}{v_{i}^{2}}{)}^{-1}$$

When adaptive weighted data fusion is combined with other unbiased estimation algorithms, it is possible to replace $M_{i}$ and $v_{i}^{2}$ in Equation (15) with an unbiased estimation value and an unbiased estimation variance. The Figure 1 shows the flowchart for batch estimation data fusion, and the flowchart for other unbiased estimation data fusion is more or less the same.

## 3. Rationale

Statistical inference is a statistical method of inferring the whole through samples, and there are two kinds of errors in statistical inference: systematic error and random error. Unbiased estimation has no systematic error because of its unbiased nature, but there is still random error, so its estimation results are not necessarily reliable.

The optimal weighting factor is calculated based on the measurement variance alone, but a smaller measurement variance does not mean a smaller measurement error. Although the measurement variance of the data measured by different sensors varies, the mathematical expectations are all equal, so, in practice, it is easy to have a smaller measurement variance but a larger measurement error.

### 3.1. Biased Estimators

**Theorem** **1.**
*The error of the unbiased estimator can be further optimized.*


**Proof** **of** **Theorem** **1.**Assuming that the true value of the measured unknown quantity x is μ, the sensor measurement data follows $N\left(\mu ,v^{2}\right)$. E is an unbiased estimator constructed using a linear combination of the sensor measurement data, and letting the estimation variance of E be $\sigma^{2}$, then $E\sim N\left(\mu ,\sigma^{2}\right)$, defining the estimation error est as follows:(17)est=E−μ$E\left(est\right)=0,D\left(est\right)=\sigma^{2}$ is easy to obtain, and est follows $N\left(0,\sigma^{2}\right)$. Assuming that the estimation error is tolerable when $est\in \left[-z,z\right]$, the probability that the estimation error is tolerable is:(18)$$P\left(est\in \left[-z,z\right]\right)=2\int_{0}^{z}\frac{1}{\sqrt{2\pi }\sigma }e^{-\frac{x^{2}}{2\sigma^{2}}}dx$$If z=σ, we obtain: $P\left(est\in \left[-\sigma ,\sigma \right]\right)\approx 68.3\%$, which means that for any σ, there is about a 31.7% chance that est is larger than the estimation standard deviation σ, so that even if σ is very small, est may be able to be reduced further. □

**Theorem** **2.**
*A biased estimation value with a smaller error can be constructed from an unbiased estimation value.*


**Proof** **of** **Theorem** **2.**Assuming unbiased estimator A follows $N\left(\mu ,\sigma^{2}\right)$, the biased estimator corresponding to A is defined as B and B=A+s, where s is called offset, which is a nonzero variable. The following results can be obtained:(19)$$E\left(B\right)=E\left(A\right)+E\left(s\right)=\mu +s\;(s\neq 0)$$Because $E\left(B\right)$ is not equal to the true value μ, B is a biased estimator.Assuming that $A_{1}$ is a specific estimate of A, its estimation error is $d_{1}$ and $d_{1}=\left|A_{1}-\mu \right|$. $s_{1}$ is the offset corresponding to $A_{1}$ and $s_{1}\propto d_{1}$, and the biased estimator corresponding to $A_{1}$ is $B_{1}$ and $B_{1}=A_{1}+s_{1}$. The estimation error of $B_{1}$ is $e_{1}$ and $e_{1}=\left|A_{1}-\mu +s_{1}\right|$, so:(20)$$e_{1}=\left\{\begin{array}{l} d_{1}+\left|s_{1}\right|\;\;\;\;if\;\left(A_{1}-\mu \right)s_{1}>0\\ d_{1}-\left|s_{1}\right|\;\;\;\;if\;\left(A_{1}-\mu \right)s_{1}<0\;and\;\left|s_{1}\right|<d_{1}\;\\ \left|s_{1}\right|-d_{1}\;\;\;\;if\;\left(A_{1}-\mu \right)s_{1}<0\;and\;\left|s_{1}\right|>d_{1} \end{array}\right.$$So, when $\left(A_{1}-\mu \right)s_{1}<0$ and $\left|s_{1}\right|<2d_{1}$, $e_{1}<d_{1}$. In other words, when the sign of $s_{1}$ is opposite to $\left(A_{1}-\mu \right)$ and $\left|s_{1}\right|<2\;*\;d_{1}$, the estimation error of biased estimation is smaller than that of unbiased estimation. □

### 3.2. Optimal Weighting Factors

**Theorem** **3.**
*In some cases, the optimal weighting factor is not optimal.*


**Proof** **of** **Theorem** **3.**Suppose there are two sensors $S_{1}$ and $S_{2}$ that simultaneously measure the unknown quantity x (the true value is μ). Their measurement variances are $v_{1}^{2}$ and $v_{2}^{2}$, and they satisfy $v_{1}^{2}<v_{2}^{2}$. Their measurement data are $z_{1}$ and $z_{2}$. For adaptive weighted data fusion, $w_{1}$, the weights of $S_{1}$, will always be greater than $w_{2}$, the weights of $S_{2}$, since $v_{1}^{2}$ < $v_{2}^{2}$.Define $m_{i}$, the measurement error (unlike est, $m_{i}$ only considers the magnitude of the error, not the sign of the error), as follows:(21)$$m_{i}=\left|z_{i}-\mu \right|$$So $m_{1}=\left|z_{1}-\mu \right|$, $m_{2}=\left|z_{2}-\mu \right|$. If $m_{1}>m_{2}$, then $w_{1}\;*\;m_{1}+w_{2}\;*\;m_{2}>w_{1}\;*\;m_{2}+w_{2}\;*\;m_{1}$, which implies that the estimation error of the fusion estimation value computed from optimal weighting factors is not minimal. So, when $v_{1}^{2}$ < $v_{2}^{2}$ but $m_{1}>m_{2}$, the optimal weighting factor is not optimal, and the probability is given as follows:(22)$$P\left(m_{1}>m_{2}\right)=2\int_{-\infty }^{+\infty }\int_{x}^{z_{1}}\frac{1}{\sqrt{2\pi }v_{2}}e^{-\frac{x^{2}}{2v_{2}^{2}}}dx\;\;dz_{1}$$It can be shown that $P\left(m_{1}>m_{2}\right)>0$, which indicates that the weights of any two sensors may not be optimal. The above proof uses measured values and measured variances, but replacing them with estimated values and estimated variances is equally valid. □

**Theorem** **4.**
*It is optimal to compute the optimal weighting factor using the measurement error.*


**Proof** **of** **Theorem** **4.**Suppose there are N sensors measuring the unknown quantity x (the true value is μ), their measurement values are $M_{1},M_{2}\cdots M_{n}$, and their measurement errors are $m_{1},m_{2}\cdots m_{n}$; moreover, there are ${0<m}_{i}<m_{i+1}$. The optimal weighting factor calculated from the measurement error is $w_{1},w_{2}\cdots w_{n}$, because $m_{i}<m_{i+1}$, then $w_{i}>w_{i+1}$, so the corresponding fusion estimation value ${\hat{M}}_{1}$ is as follows:(23)$${\hat{M}}_{1}=\sum_{i=1}^{N}w_{i}M_{i}$$In this paper, only one case is proved, which has two weights different from $w_{1},w_{2}\cdots w_{n}$, and the proof can be easily generalized to other cases.Counterfactual: suppose that when the optimal weighting factor is $v_{1},v_{2},w_{3}\cdots w_{n}$ and $v_{1}<v_{2}$, the fusion estimation value ${\hat{M}}_{2}$ is more accurate than ${\hat{M}}_{1}$, where ${\hat{M}}_{2}$ is given as follows:(24)$${\hat{M}}_{2}=v_{1}M_{1}+v_{2}M_{2}+\sum_{i=3}^{N}w_{i}M_{i}$$The common term of ${\hat{M}}_{1}$ and ${\hat{M}}_{2}$ is ${\hat{M}}_{3}$, and ${\hat{M}}_{3}=\sum_{i=3}^{N}w_{i}M_{i}$, $E({\hat{M}}_{3})=\alpha \mu$ where $\alpha =\sum_{i=3}^{N}w_{i}$. So ${\hat{M}}_{1}$ and ${\hat{M}}_{2}$ become this form:(25)$${\hat{M}}_{1}=w_{1}M_{1}+w_{2}M_{2}+{\hat{M}}_{3},\;{\hat{M}}_{2}=v_{1}M_{1}+v_{2}M_{2}+{\hat{M}}_{3}$$Because $w_{1}+w_{2}\cdots +w_{n}=v_{1}+v_{2}+w_{3}\cdots +w_{n}=1$, so $w_{1}+w_{2}=v_{1}+v_{2}$ and set $w_{1}+w_{2}=\theta$. Substituting $M_{i}=\mu +m_{i}$ for ${\hat{M}}_{1}$ and ${\hat{M}}_{2}$, then we get the following equation:(26)$${\hat{M}}_{1}=\left(w_{1}+w_{2}\right)\mu +w_{1}m_{1}+w_{2}m_{2}+{\hat{M}}_{3}$$
(27)$${\hat{M}}_{2}=\left(v_{1}+v_{2}\right)\mu +v_{1}m_{1}+v_{2}m_{2}+{\hat{M}}_{3}\;=\left(w_{1}+w_{2}\right)\mu +v_{1}m_{1}+v_{2}m_{2}+{\hat{M}}_{3}$$Because ${\hat{M}}_{3}+\left(w_{1}+w_{2}\right)\mu =\mu$, so the larger $\left|w_{1}m_{1}+w_{2}m_{2}\right|$ and $\left|v_{1}m_{1}+v_{2}m_{2}\right|$ are, the less accurate ${\hat{M}}_{1}$ and ${\hat{M}}_{2}$ are. Because $m_{1}>0$ and $m_{2}>0$, so $\left|w_{1}m_{1}+w_{2}m_{2}\right|$ = $w_{1}m_{1}+w_{2}m_{2}$ and $\left|v_{1}m_{1}+v_{2}m_{2}\right|$ = $v_{1}m_{1}+v_{2}m_{2}$. Set $f\left(\gamma \right)$ is given as follows:(28)$$f\left(\gamma \right)=\gamma m_{1}+(\theta -\gamma )m_{2}$$So $f\left(w_{1}\right)=w_{1}m_{1}+w_{2}m_{2}$ and $f\left(v_{1}\right)=v_{1}m_{1}+v_{2}m_{2}$. The derivative of $f\left(\gamma \right)$ is given as follows:(29)$$\frac{df\left(\gamma \right)}{d\gamma }=m_{1}-m_{2}<0$$So, the larger γ is, the smaller $f\left(\gamma \right)$. Because $w_{1}>w_{2}$ and $v_{1}<v_{2}$, so $w_{1}>\frac{\theta }{2}$, $v_{1}<\frac{\theta }{2}$, and $w_{1}m_{1}+w_{2}m_{2}<v_{1}m_{1}+v_{2}m_{2}$, ${\hat{M}}_{1}$ is more accurate than ${\hat{M}}_{2}$. Contradicting the hypothesis, the optimal weighting factor calculated from the estimation error is better. The above proof uses measured values and measured variances, but replacing them with estimated values and estimated variances is equally valid. □

## 4. Algorithm Implementation

The algorithm in Section 4.1 only uses measurement data from a single sensor, so it can be implemented in a distributed manner. The algorithms in Section 4.2 and Section 4.3 include data fusion algorithms, so they can only be implemented using centralized methods.

### 4.1. Unbiased Estimation

Suppose that N sensors simultaneously measure the unknown quantity x (the true value is *µ*), and all the sensor nodes in a network measure the observed physical quantity at the same time. For any of the sensors, if it collects 10 data over a period of time, in order from smallest to largest: $a_{1}{,a}_{2},\cdots {,a}_{10}$, then construct the unbiased estimators as follows:(30)$$E_{1}=\frac{a_{5}+a_{6}}{2},\;E_{2}=\frac{a_{1}+a_{2}+a_{9}+a_{10}}{4},\;E_{3}=\frac{a_{3}+a_{4}+a_{7}+a_{8}}{4},\;E_{4}=\frac{1}{10}\sum_{i=1}^{10}a_{i}$$

Assuming that the sensor follows $N\left(\mu ,\sigma^{2}\right)$, theoretically, $E_{1}\sim N\left(\mu ,\frac{\sigma^{2}}{2}\right)$ and $E_{2},E_{3}\sim N\left(\mu ,\frac{\sigma^{2}}{4}\right)$. Because $a_{1}$~$a_{10}$ are data sorted in order of size, these estimators have lower estimated variances. Through experiments, it was found that the estimated variances of $E_{1}\sim E_{3}$ are approximately $\frac{\sigma^{2}}{7.5},\frac{\sigma^{2}}{8},\frac{\sigma^{2}}{9}$. Therefore, $E_{1}\sim E_{3}$ is independent of each other and has an estimated variance smaller than the measured variance.

$\hat{E}$, the mean of estimation values, and $\sigma_{i}^{2}$, the estimation variance, are given as follows:(31)$$\hat{E}=\frac{\sum_{i=1}^{4}E_{i}}{4},\;\sigma_{i}^{2}={\left(E_{i}-\hat{E}\right)}^{2}$$

An adaptive weighted data fusion algorithm is used to obtain $\hat{x}$, the unbiased estimation value, and $\hat{\sigma^{2}}$, the unbiased estimation variance. And the specific formula is as follows:(32)$$w_{i}=(\sigma_{i}^{2}\sum_{i=1}^{n}\frac{1}{\sigma_{i}^{2}}{)}^{-1},\hat{x}=\sum_{i=1}^{4}w_{i}E_{i},\;{\hat{\sigma }}^{2}=(\sum_{i=1}^{n}\frac{1}{\sigma_{i}^{2}}{)}^{-1}$$

Because $E_{4}$ is the least squares estimator, theoretically, the accuracy of the fused estimate $\hat{x}$ will be higher than that of the least squares estimate.

If the number of measurement data collected by a sensor within a certain period is m and m is not equal to 10, the unbiased estimator can be constructed according to the following rules:Each group consists of at least two measurement values, and each group does not use duplicate measurement values. The unbiased estimator corresponding to each group is the mean of the measurement values within the group.When m is even (m = 2k), for any measurement value $a_{i}$, it is necessary to ensure that $a_{i}$ and $a_{m-i}$ are in the same group.When m is odd (m = 2k + 1), for any measurement value $a_{i}$, it is necessary to ensure that $a_{i}$ and $a_{m-i}$ are in the same group. In addition, it is also necessary to ensure that $a_{k}$, $a_{k+1}$, and $a_{k+2}$ are in the same group.Add an additional group that includes all measured values, such as $E_{4}$ in Equation (30).

### 4.2. Biased Estimation

$\left|s_{i}\right|$, the size of the offset $s_{i}$, needs to be determined first. For the *i*th sensor, define its unbiased estimation value and unbiased estimation variance as $A_{i}$ and $\sigma_{i}^{2}$. The fusion estimation value $\hat{A}$ and the fusion estimation variance of these unbiased estimation values ${\hat{\sigma }}^{2}$ are computed by adaptive weighted data fusion algorithm.

Define $g_{i}$ as the degree of deviation of the *i*th unbiased estimation value from the true value. And $g_{i}$ is given as (use $\hat{A}$ replaces true value) follows:(33)$$g_{i}=\frac{\left|A_{i}-\hat{A}\right|}{\hat{\;\sigma }}$$

Define ω as the coefficient of $\left|s_{i}\right|$, where the size of $s_{i}$, and the value of $\omega_{i}$ are determined by $g_{i}$. When $g_{i}<0.1$, $\omega_{i}=0$, when ${1>g}_{i}>0.1$, $\omega_{i}=0.5$. Otherwise, $\omega_{i}=1$. ω is used to control the size of the offset and avoid excessive offset.

$\left|s_{i}\right|$ is given as follows:(34)$$\left|s_{i}\right|=\frac{\sigma_{i}}{\sqrt{N}}\cdot g_{i}\cdot \omega_{i}$$

Then, the sign of $s_{i}$ needs to be determined. $s_{i}$ takes a negative sign when $A_{i}$ > $\hat{A}$ and a positive sign otherwise. So $B_{i}$ is given as follows:(35)$$\;B_{i}=\left\{\begin{array}{c} \;\frac{\sigma_{i}}{\hat{\sigma }\;*\;\sqrt{N}}\cdot \;g_{i}\cdot \hat{\sigma }\cdot \omega_{i}\;\;\;\;\;if\;A_{i}>\hat{A}\;\\ -\frac{\sigma_{i}}{\hat{\sigma }\;*\;\sqrt{N}}\cdot \;g_{i}\cdot \hat{\sigma }\cdot \omega_{i}\;\;\;if\;A_{i}<\hat{A} \end{array}\right.$$

### 4.3. Data Fusion

The application of the adaptive weighted data fusion algorithm requires measuring variance, which is replaced by estimated variance in unbiased estimation data fusion. In biased estimation data fusion, the estimated variance is calculated based on the estimation error, and the design concept of the calculation formula comes from the Kalman gain. Define $e_{i}$ as the biased estimation error of $B_{i}$, and $e_{i}$ is given as follows:(36)$$e_{i}=B_{i}-\mu$$

The true value is unknown; use $\hat{A}$ instead of the true value. So $e_{i}^{2}$ is given as:(37)$$e_{i}^{2}={\left(B_{i}-\hat{A}\right)}^{2}$$

Then, through 100,000 experiments, calculate ${\bar{e}}_{i}^{2}$, mean of $e_{i}^{2}$, and ${\bar{g}}_{i}$, mean of $g_{i}$, for different g-value intervals. The ${\bar{e}}_{i}^{2}$ and ${\bar{g}}_{i}$ over the three g intervals are shown in the Table 1.

${\bar{e}}_{i}^{2}$ and ${\bar{g}}_{i}$ can be represented as follows:(38)$$\left\{\begin{array}{c} {\bar{e}}_{i}^{2}=0.0176,{\bar{g}}_{i}=0.0493\;\;\;\;\;\;\;\;\;if\;g_{i}<0.1\;\;\\ {\bar{e}}_{i}^{2}=0.0203,{\bar{g}}_{i}=0.5268\;\;\;\;if\;0.1\leq g_{i}<1\\ {\bar{e}}_{i}^{2}=0.0293,{\bar{g}}_{i}=1.6735\;\;\;\;\;\;\;\;\;\;\;\;\;\;\;if\;g_{i}\geq 1 \end{array}\right.$$

So, ${\tilde{\sigma }}_{i}^{2}$, which represents the biased estimation variance of $B_{i}$, is given as follows:(39)$${\tilde{\sigma }}_{i}^{2}=e_{i}^{2}+\frac{g_{i}}{g_{i}+{\bar{g}}_{i}}\cdot ({\bar{e}}_{i}^{2}-e_{i}^{2})$$

Thus, $w_{i}$, the optimal weighting factor, and $\hat{M}$, the fusion estimation value, are given as follows:(40)$$w_{i}=({\tilde{\sigma }}_{i}^{2}\sum_{i=1}^{n}\frac{1}{{\tilde{\sigma }}_{i}^{2}}{)}^{-1},\;\hat{M}=\sum_{i=1}^{N}w_{i}B_{i}$$

The flowchart of biased estimation data fusion is shown in Figure 2:

## 5. Results

The simulation test is divided into two parts as follows: 1. a normal distribution white noise test (all noise follows a normal distribution), and 2. A uniform distribution white noise test (environmental noise is uniformly distributed white noise). The test items for the two tests are shown in Table 2:

The testing mode is as follows: ten rounds of experiments are conducted for each combination of test items, and ten tests are conducted for each round. The average of the ten test results is used as the result of this round of experiments. Three sensors are used for each test, with 10 measurement data used for each sensor.

The test indicators are mean relative error (mre) and mean square error (mse). mre is used to compare the average performance of the three algorithms, and mse is used to compare the stability of the three algorithms. Their calculation formulas are as follows:(41)$${mre}_{i}=\frac{\sum_{j=1}^{10}\frac{\left(m_{ij}-\mu \right)}{\mu }}{10}$$
(42)$${mse}_{i}=\frac{\sum_{j=1}^{10}{\left(m_{ij}-\mu \right)}^{2}}{10}$$
where i represents the number of test rounds and $m_{ij}$ represents the estimated value of the jth test in the *i*th round. Relative improvement (ri) is used to compare the performance differences between two algorithms, and the calculation formula for ri is as follows:(43)$$ri=\frac{V_{\alpha }-V_{\beta }}{V_{\alpha }}\times 100\%$$
where $V_{\alpha }$ and $V_{\beta }$ represent the same indicator value (e.g., mre) for algorithms α and algorithms β. And, if the indicator has multiple values, calculate the mean of the indicator first, and then calculate ri. If ri is greater than 0, it means that algorithm β has improved ri compared to algorithm α.

### 5.1. Normal Distribution White Noise Test

The test results are as follows (Figure 3, Figure 4 and Figure 5 show the test results when the measurement noise is 0.25, and the remaining result graphs are Figure A1, Figure A2, Figure A3, Figure A4, Figure A5 and Figure A6 in Appendix A):

From Table 3, it can be seen that the mre of biased estimation data fusion (BED) decreased by an average of 9.25% and 10.26% compared to least squares estimation data fusion (LSD) and batch estimation data fusion (BTD). Therefore, BED has higher fusion accuracy.

From Table 4, it can be seen that the mse of BED decreased by an average of 16.24% and 20.28% compared to LSD and BTD. Therefore, the bed has higher stability.

And, as the noise variance increases, the average increases in mre and mse of LSD are 0.297% and 0.036, the average increases in mre and mse of BTD are 0.306% and 0.038, and the average increases in the mre and mse of BED are 0.269% and 0.03. So, mre and mse of BED decreased by an average of 11.59% and 18.85% compared to LSD and BTD, respectively. Therefore, BED has higher noise resistance.

### 5.2. Uniform Distribution White Noise Test

The test results are as follows (Figure 6, Figure 7 and Figure 8 show the test results when the measurement noise is 0.25, and the remaining result graphs are shown in Figure A7, Figure A8, Figure A9, Figure A10, Figure A11 and Figure A12):

From Table 5, it can be seen that the mre of BED decreased by an average of 8.56% and 6.58% compared to LSD and BTD. Therefore, BED has higher fusion accuracy, but the advantage of mre is reduced by 22.39% compared to normal distribution white noise testing.

From Table 6, it can be seen that the mse of BED decreased by an average of 17.42% and 21.01% compared to LSD and BTD. Therefore, BED has higher fusion accuracy, but the advantage of mse is increased by 5.23% compared to normal distribution white noise testing.

As the noise variance increases, the average increases in mre and mse of LSD are 0.305% and 0.03, the average increases in mre and mse of BTD are 0.298% and 0.031, and the average increases in mre and mse of BED are 0.278% and 0.024. So, mre and mse of BED decreased by an average of 8.21% and 21.29% compared to LSD and BTD. BED has higher noise resistance, and the comprehensive noise resistance is equivalent to that of normal distribution white noise testing.

In summary, even if the noise does not conform to a normal distribution, as long as the mean noise is 0, the algorithm can perform well.

## 6. Discussion

In this paper, an adaptive weighting data fusion algorithm based on biased estimation is proposed to address the shortcomings of unbiased estimation data fusion. The algorithm in this paper first further reduces the estimation error by biased estimation and then uses the estimation error instead of the measurement variance to compute the optimal weighting factor, which avoids the fact that the optimal weighting factor is not optimal in some cases. The final results of the simulation experiments show that this algorithm outperforms the original algorithm in terms of fusion accuracy, stability, and noise resistance, which are better than other algorithms.

The essence of this algorithm is to sacrifice unbiasedness for higher estimation accuracy, so the overall accuracy of biased estimation is significantly higher than that of unbiased estimation. However, due to the loss of unbiasedness, the performance of the algorithm is not as good as expected when using adaptive weighted data fusion, so there is still much room for improvement.

## Figures and Tables

**Figure 1 sensors-24-03275-f001:**
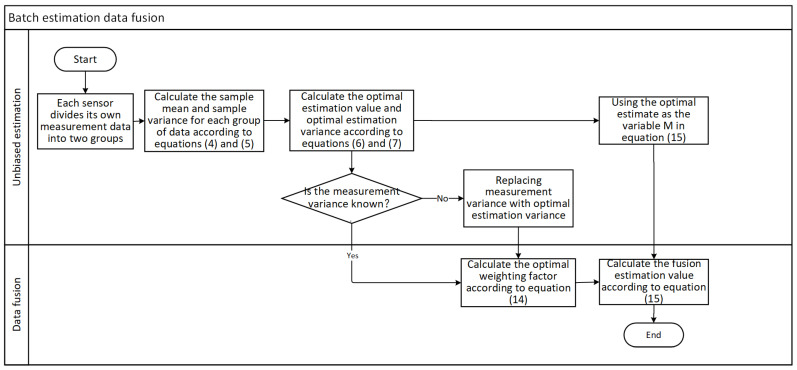
Flowchart of batch estimation data fusion.

**Figure 2 sensors-24-03275-f002:**
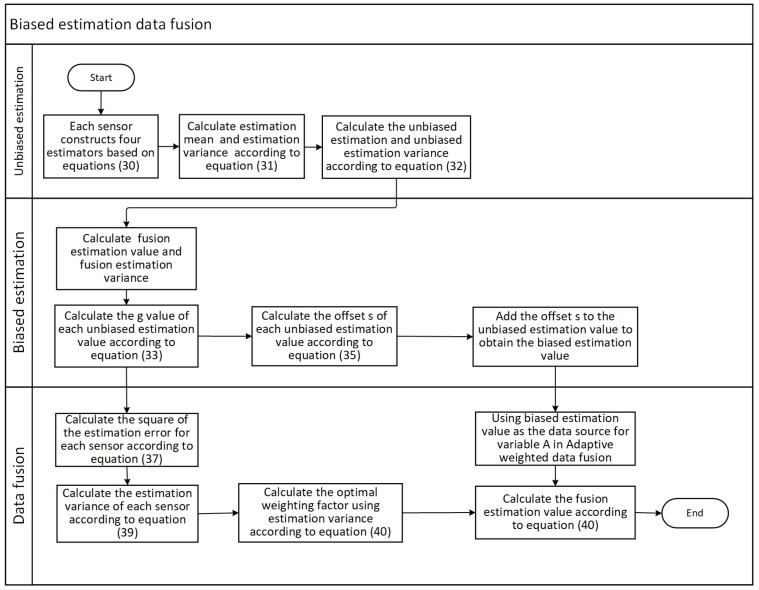
Flowchart of biased estimation data fusion.

**Figure 3 sensors-24-03275-f003:**
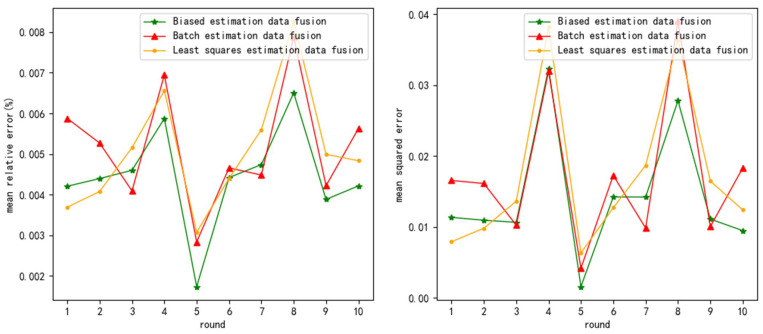
mre and mse when measurement variance = 0.25 and normal distribution noise variance = 0.2.

**Figure 4 sensors-24-03275-f004:**
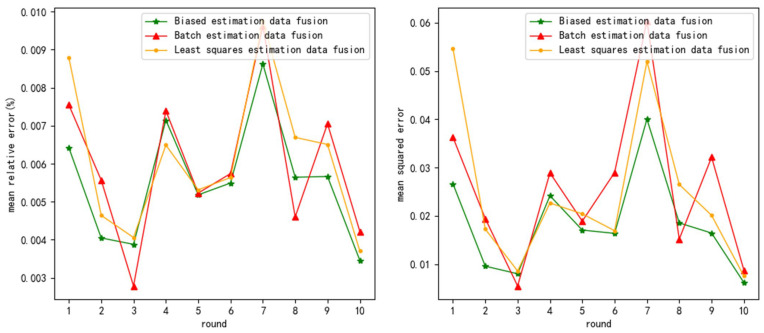
mre and mse when measurement variance = 0.25 and normal distribution noise variance = 0.5.

**Figure 5 sensors-24-03275-f005:**
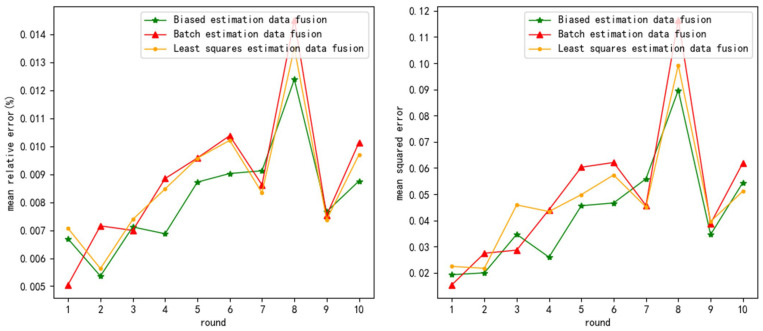
mre and mse when measurement variance = 0.25 and normal distribution noise variance = 1.

**Figure 6 sensors-24-03275-f006:**
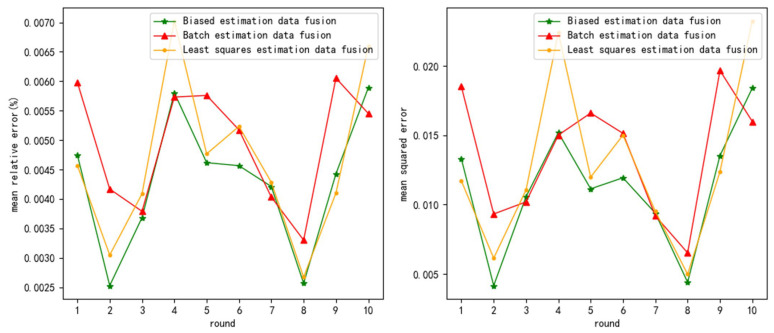
mre and mse when measurement variance = 0.25 and uniformly distributed noise variance = 0.2.

**Figure 7 sensors-24-03275-f007:**
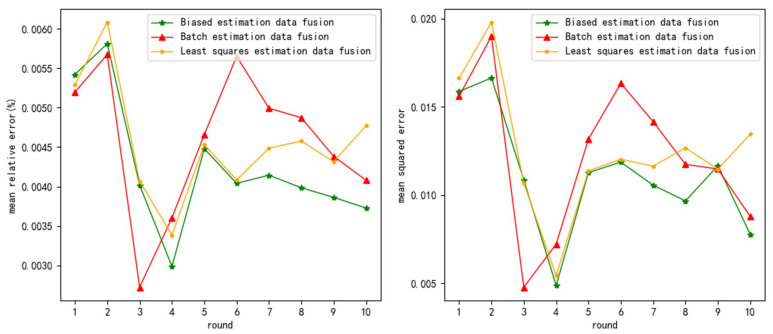
mre and mse when measurement variance = 0.25 and uniformly distributed noise variance = 0.5.

**Figure 8 sensors-24-03275-f008:**
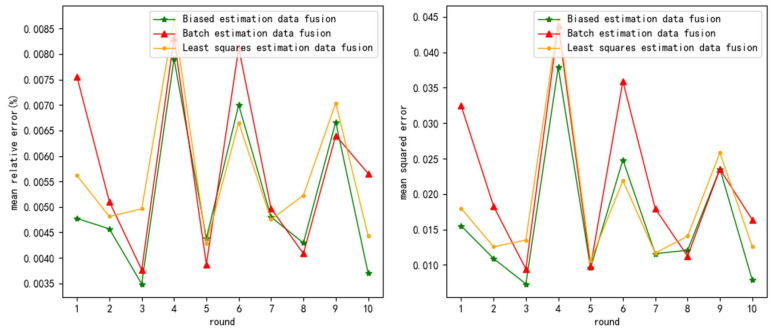
mre and mse when measurement variance = 0.25 and uniformly distributed noise variance = 1.

**Table 1 sensors-24-03275-t001:** Mean estimation error and mean g for different g intervals.

$g_{i}$	${\bar{e}}_{i}^{2}$	${\bar{g}}_{i}$
$g_{i}<0.1$	0.0176	0.0493
$0.1\leq g_{i}<1$	0.0203	0.5268
$g_{i}\geq 1$	0.0293	1.6735

**Table 2 sensors-24-03275-t002:** The test items for the two tests.

Test Items	Test Value 1	Test Value 2	Test Value 3
True value	20	null	null
Measurement variance	0.25	1	2
Noise variance	0.2	0.5	1

**Table 3 sensors-24-03275-t003:** mre of all algorithms when noise follows normal distribution.

Measure Variance	Noise Variance	Least Squares Estimation Data Fusion	Batch Estimation Data Fusion	Biased Estimation Data Fusion
0.25	0.2	0.506%	0.518%	0.445%
	0.5	0.615%	0.597%	0.555%
	1.0	0.872%	0.887%	0.816%
	mean	0.664%	0.667%	0.605%
1	0.2	1.048%	1.043%	0.943%
	0.5	1.131%	1.137%	1.004%
	1.0	1.035%	1.053%	0.958%
	mean	1.071%	1.077%	0.968%
2	0.2	1.018%	1.022%	0.870%
	0.5	1.309%	1.345%	1.222%
	1.0	1.449%	1.471%	1.339%
	mean	1.258%	1.279%	1.143%

**Table 4 sensors-24-03275-t004:** mse of all algorithms when noise follows normal distribution.

Measure Variance	Noise Variance	Least Squares Estimation Data Fusion	Batch Estimation Data Fusion	Biased Estimation Data Fusion
0.25	0.2	0.017	0.017	0.014
	0.5	0.024	0.025	0.018
	1.0	0.047	0.050	0.042
	mean	0.029	0.030	0.024
1	0.2	0.068	0.067	0.053
	0.5	0.075	0.081	0.061
	1.0	0.065	0.066	0.054
	mean	0.069	0.071	0.056
2	0.2	0.073	0.079	0.055
	0.5	0.103	0.104	0.091
	1.0	0.127	0.136	0.111
	mean	0.101	0.106	0.085

**Table 5 sensors-24-03275-t005:** mre of all algorithms when noise follows uniform distribution.

Measure Variance	Noise Variance	Least Squares Estimation Data Fusion	Batch Estimation Data Fusion	Biased Estimation Data Fusion
0.25	0.2	0.463%	0.494%	0.430%
	0.5	0.455%	0.458%	0.424%
	1.0	0.564%	0.578%	0.518%
	mean	0.494%	0.510%	0.457%
1	0.2	0.762%	0.830%	0.705%
	0.5	0.769%	0.794%	0.685%
	1.0	0.962%	0.947%	0.865%
	mean	0.831%	0.857%	0.751%
2	0.2	1.075%	1.084%	0.981%
	0.5	1.058%	1.071%	0.982%
	1.0	1.180%	1.167%	1.078%
	mean	1.104%	1.107%	1.013%

**Table 6 sensors-24-03275-t006:** mse of all algorithms when noise follows uniform distribution.

Measure Variance	Noise Variance	Least Squares Estimation Data Fusion	Batch Estimation Data Fusion	Biased Estimation Data Fusion
0.25	0.2	0.012	0.013	0.011
	0.5	0.012	0.012	0.011
	1.0	0.018	0.021	0.016
	mean	0.014	0.015	0.012
1	0.2	0.037	0.043	0.032
	0.5	0.041	0.041	0.031
	1.0	0.054	0.055	0.046
	mean	0.044	0.046	0.036
2	0.2	0.071	0.075	0.058
	0.5	0.067	0.071	0.058
	1.0	0.085	0.086	0.069
	mean	0.074	0.077	0.061

## Data Availability

Data are contained within the article.

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
