# Peer review of "Multi-Sensor Adaptive Weighted Data Fusion Based on Biased Estimation"

_sensors, 2024, doi:10.3390/s24113275_

Round 1

Reviewer 1 Report

Comments and Suggestions for Authors

This paper proposed a multi-sensor adaptive weighted data fusion algorithm based on biased estimation

.

The following comments must be addressed before publication.

1. line 45: "Adaptive weighted data fusion is a widely used multi-sensor data fusion algorithm" lacks references

2. line 84: "in practice the measurement variance is often unknown.", please rewrite this sentence as the measurement covariance can be obtained by experiments.

3. The coefficients of Eq. (30) should be explained clearly. It is confusing where they come from.

4. line 207: "the sensor collects 10 data over a period of time", what if the number of measurements is not 10? for example 9 or 11, does the equations still hold in these cases?

Comments on the Quality of English Language

English is well

Author Response

  1. line 45: "Adaptive weighted data fusion is a widely used multi-sensor data fusion algorithm" lacks references

Quotes have been added to the original text

  1. line 84: "in practice the measurement variance is often unknown.", please rewrite this sentence as the measurement covariance can be obtained by experiments.

The sentences in the original text have been modified.

  1. The coefficients of Eq. (30) should be explained clearly. It is confusing where they come from.

Further explanations have been added to the original text, as follows:

Suppose that N sensors simultaneously measure the unknown quantity  (the true value is µ). For any of the sensors, if it collects 10 data over a period of time, in order from smallest to largest: , then construct the unbiased estimators as:

, , ,

Assuming that the sensor follows , so theoretically,and . Because ~ are data sorted in order of size, these estimators have lower estimated variances, through experiments, it was found that the estimated variances of  are approximately , Therefore,  is independent of each other and has an estimated variance smaller than the measured variance.

In summary, this is to construct an unbiased estimator with small estimation variance, in order to make the fused estimation values  as accurate as possible. The relevant calculation formulas in the original text are as follows:

, the mean of estimation values, and , the estimation variance, are given as:

,

An adaptive weighted data fusion algorithm is used to obtain , unbiased estimation value, and , unbiased estimation variance. And the specific formula is as follows:

  1. line 207: "the sensor collects 10 data over a period of time", what if the number of measurements is not 10? for example 9 or 11, does the equations still hold in these cases?

Further explanations have been added to the original text, as follows:

If the number of data points collected by a sensor within a certain period is m and m is not equal to 10, the unbiased estimator can be constructed according to the following rules:

1.Each group consists of at least two measurement values, and each group does not use duplicate measurement values. The unbiased estimator corresponding to each group is the mean of the measurement values within the group.

2.When m is even (m=2k), for any measurement value , it is necessary to ensure that  and  are in the same group.

3.When m is odd (m=2k+1), for any measurement value , it is necessary to ensure that  and  are in the same group. In addition, it is also necessary to ensure that ,  and  are in the same group.

4.Add an additional group that includes all measured values, such as  in Eq. (30).

Reviewer 2 Report

Comments and Suggestions for Authors

please see the attached pdf file

Author Response

  1. Line 36: It is stated: "... which can improve the fusion accuracy because the estimation value has higher accuracy than the measurement value." Is it possible to quantify (for example, in percentages) how data fusion can increase accuracy? What value is expected in real-world applications?

Assuming the true value is , the measurement variance of a sensor is , and the sensor has  independent measurement data, namely: , then these data follow , using least squares estimation as an example, construct the following unbiased estimator:

Then:

Hypothesis , so the normal distribution curve between the measured value and the unbiased estimator is as follows:

Comparing the area enclosed by the two curves and the x-axis, it can be found that the unbiased estimator occupies more area around , indicating that the unbiased estimator is generally closer to the true value.

  1. Line 48: I am interested in how "priori knowledge of the measurement variance" can be determined.

There are two methods:

  1. First, use this sensor to measure the same object and obtain a large amount of measurement data. Then, calculate the sample mean and sample variance, and treat the sample variance as the measurement variance of the sensor. The formula is as follows:

This approach cannot detect whether the mean of measurement noise or environmental noise is zero, so there may be bias.

  1. First, use a high-precision measuring instrument to measure the true value of the object in advance (recorded as μ), Then use sensors to measure the object and obtain a large amount of measurement data, and then treat μ as the mean, calculate the sample variance. The formula is as follows:

Because the true mean is known, it is n instead of n-1 (formula source: https://www.visiondummy.com/2014/03/divide-variance-n-1/).

This approach cannot detect whether the mean of measurement noise or environmental noise is , if , each measurement result can be corrected to make the mean of noise become zero.

Obviously, for sensors, this approach is not possible in the vast majority of cases. In addition, the following assumption is generally adopted: the electronic noise (measurement noise) and environmental noise inside electronic instruments are both zero mean white noise, and Kalman filtering is also based on this assumption. Therefore, using method one is a common approach.

  1. In the Introduction, I recommend adding a more extensive insight into data fusion. I recommend mentioning that the application of data fusion algorithms can suppress negative environmental factors (e.g., radiation, noise, temperature, weather conditions, etc.) that worsen the precision of measurement values in many technologies. Could you cite these papers [ https://doi.org/10.3390/s20133677

,  https://doi.org/10.1016/j.compstruct.2019.111798 ] where this issue is comprehensively analyzed in real-world scenarios?

I have added relevant content in the introduction and cited relevant literature.

The application of data fusion algorithms can suppress negative environmental factors, in practical applications, it is mainly reflected in: 1. High error sensors can also play a role in reducing algorithm errors. 2. Reduce the impact of environmental noise through data from multiple information sources. 3. The stability of the measurement results of a single sensor is poor, and data fusion improves the stability of the results.

  1. Line 75 - It is stated: "?(?)? is the measurement value of the ith sensor at the moment k". I recommend specifying whether you assume that all the sensor nodes in a network measure the observed physical quantity at the same time.

It has been specified in the text that all sensors collect data at the same time.

  1. Line 100 - It is stated: "these n data are equally divided into 2 groups". I recommend specifying criteria for how the data is divided.

It has been specified in the text that these n data will be randomly divided into two groups.

  1. Does your approach work in a fully distributed way? Is any centralization required? This should be specified in the paper.

The algorithm in section 4.1 only uses measurement data from a single sensor, so it can be implemented in a distributed manner. The algorithms in sections 4.2 and 4.3 include data fusion algorithms, so they can only be implemented using centralized methods.

The above content has been noted in the text, with specific locations at the beginning of Chapter 4.

  1. Line 207: It is stated: „the measurement data of one of sensors obeys ?(μ, σ2)“. Is this statement confirmed by any real-world measurement (for example, in another paper)?

Or why do you consider this distribution of the measurement data?

In papers, books, and videos introducing the principle of adaptive weighted data fusion, as well as explanations of the Kalman filter, it is commonly assumed that electronic noise (measurement noise) within electronic instruments and environmental noise are both zero-mean white noise.

Assuming that the true value is μ, and the measurement noise and environmental noise follow  and  respectively, according to the additivity of normal distributions and the calculation formulas for expectation and variance, it can be deduced that the sensor obeys ?(μ, +), which can be abbreviated as .

The name of the book is "Optimal State Estimation," which primarily covers the relevant knowledge of the Kalman filter.

  1. Line 208: It is stated: "in order from smallest to largest". Is this statement valid also if the true value can change over time?

If the true value changes over time, it is no longer valid because these measurement data are no longer independent and identically distributed, and Kalman filtering is required.

  1. In the experimental part, errors in the figures sometimes grow when the number of rounds increases. Usually, more rounds ensure that the value of errors is decreased. Is it possible to justify it?

The testing mode is: ten rounds of experiments will be conducted for each combination of test items, and ten tests are conducted for each round. The average of the ten test results will be used as the result of this round of experiments. Three sensors are used for each test, with 10 measurement data used for each sensor.

Therefore, the amount of data will not increase with the increase of testing times, and adopting this testing mode is to ensure that the test results do not change due to the amount of data.

In practice, as the amount of data increases, the error does indeed decrease. For example, the variance of the least squares estimation is equal to the variance of the measurement data divided by the amount of data.

  1. Some formal mistakes have been detected:
    • Some symbols are written in italic font, while others are not. For example: Line 74 - x is italic, μ is not. I recommend unifying it.
    • Line 114: I assume that IS is missing here: "the auxiliary function that constructed". In my opinion, the correct version should be "the auxiliary function that is constructed" OR "the auxiliary function constructed" This part needs revision.
    • Line 180 - 189: a non-common symbol for a comma "、" is used several times.

I recommend checking whether or not it was used intentionally or if it is a typo. 

  • For example, in Formula (35), you used „“. It is not explained what this symbol represents. Does this symbol represent MULTIPLICATION? Because in for example (26), this symbol is not used to express multiplication.

I have corrected the relevant errors in the text and now use italics for all variables. I have checked them again. Thank you very much for your correction.

Round 2

Reviewer 1 Report

Comments and Suggestions for Authors

I have no more comments. Thank you.

Reviewer 2 Report

Comments and Suggestions for Authors The authors precisely addressed my recommendations. In my opinion, the paper can be accepted in its current form.